# Effects of β-Alanine Supplementation on Subjects Performing High-Intensity Functional Training

**DOI:** 10.3390/nu16142340

**Published:** 2024-07-19

**Authors:** Eduardo Cimadevilla-Fernández-Pola, Cristina Martínez-Roldán, Jose Luis Maté-Muñoz, Jesús Guodemar-Pérez, Maria Aránzazu Sánchez-Calabuig, Pablo García-Fernández, Juan Pablo Hervás-Pérez, Juan Hernández-Lougedo

**Affiliations:** 1Physiotherapy and Health Research Group (FYSA), Faculty of Health Sciences-HM Hospitals, University Camilo José Cela, Urb. Villafranca del Castillo, 49. Villanueva de la Cañada, 28692 Madrid, Spain; ecimadevilla@ucjc.edu (E.C.-F.-P.); cmroldan@ucjc.edu (C.M.-R.); jguodemar@ucjc.edu (J.G.-P.); masanchez@ucjc.edu (M.A.S.-C.); jlougedo@ucjc.edu (J.H.-L.); 2Instituto de Investigación Sanitaria HM Hospitales, 28692 Madrid, Spain; 3Faculty of Nursing, Physiotherapy and Podiatry, Complutense University of Madrid, 28040 Madrid, Spain; jmate03@ucm.es (J.L.M.-M.); pablga25@ucm.es (P.G.-F.)

**Keywords:** β-alanine, carnosine, supplementation, perceived intensity, muscle fatigue, HIFT, performance sport

## Abstract

Background: β-alanine, a non-essential amino acid found in the diet and produced through nucleotide catabolism, is significant for muscle performance due to its role in carnosine synthesis. This study aims to assess the impact of a 4-week β-alanine supplementation on neuromuscular fatigue in individuals engaging in High-Intensity Functional Training (HIFT) and its subsequent effect on sports performance, distinguishing between central fatigue from the CNS and peripheral fatigue from the muscular system. Materials and methods: This study (a randomized controlled trial) comprised a total of 27 subjects, who were divided into two groups. Group A (the control group) was administered sucrose powder, while Group B (the experimental group) was given β-alanine powder. The subjects were randomly assigned to either the experimental or control groups. This study lasted four weeks, during which both groups participated in high-intensity interval training (HIFT) on the first day to induce fatigue and work close to their VO_2_ max. Results: Statistically significant changes were in the sports performance variables, specifically vertical jump and jumping power (*p* = 0.027). These changes were observed only in the group that had been supplemented with β-alanine. Nevertheless, no alterations were observed in any other variables, including fatigue, metabolic intensity of exercise, or perceived intensity (*p* > 0.05). Conclusions: A four-week β-alanine intake program demonstrated an improvement in the capacity of subjects, as evidenced by enhanced vertical jump and power performance. Nevertheless, it does result in discernible alterations in performance.

## 1. Introduction

β-Alanine is a non-essential amino acid that is naturally present in our daily diet. Our body is also capable of synthesizing it in the liver through the catabolism of pyrimidine nucleotides. β-Alanine is particularly relevant for muscle performance due to its influence on carnosine [1,2]. Carnosine is a dipeptide that is abundantly stored in the skeletal muscle of vertebrates, including humans [3]. It is composed of the amino acids β-alanine and L-histidine. Carnosine synthesis is mediated by carnosine synthase, which requires both L-histidine and β-alanine [4]. However, muscle cannot directly absorb carnosine from the bloodstream due to the low concentrations of β-alanine in muscle compared to histidine and carnosine synthase [5,6]. Furthermore, the endogenous synthesis of β-alanine occurs primarily in small liver cells, thereby limiting the availability of dietary β-alanine for carnosine synthesis in skeletal muscle [1,7,8,9].

Carnosine plays several important roles in the human body, particularly in athletic performance [1,10,11,12,13]. It acts as an intracellular buffer, helping to maintain acid-base balance during physical activity. Carnosine acts as an antioxidant, contributing to the protection of cells against oxidative stress. It is also involved in the activation of myosin ATPase, which is responsible for the maintenance of ATP reserves [14,15,16]. Furthermore, carnosine increases the sensitivity of muscle fibers to calcium, which may result in enhanced muscle strength and reduced neuromuscular fatigue [7,17,18].

It is noteworthy that there are differences in the concentration of carnosine between muscle fiber types [19,20]. The literature indicates that fast twitch or type II fibers have approximately 1.2 times more carnosine than slow twitch or type I fibers, in addition to a higher creatine content [3,21,22]. This aligns with the composition of muscle fibers in athletes who engage in explosive sports, such as high-intensity interval training (HIIT), which tends to have a higher proportion of type II fibers and increase normal levels of testosterone synthesis [23,24]. It has been observed that β-alanine supplementation can significantly increase carnosine levels in muscle. One study demonstrated that a diet supplemented with β-alanine for 4–10 weeks, at a dose of 4.8 g per day, increased carnosine levels in cyclists by 60–80% [2]. This can help delay fatigue and improve performance. However, to achieve a significant increase in muscle carnosine reserves, a minimum period of 2 weeks at a dose of 1.6 g per day is required [25]. It is important to note that any excess β-alanine and carnosine are excreted in the urine [1].

High-Intensity Functional Training (HIFT) combines a variety of exercises to promote a healthy and functional lifestyle. These include functional and natural movements such as strength, weightlifting, cardiovascular, and plyometric exercises. The objective is to enhance strength, hypertrophy, fat oxidation, muscular endurance, and optimize body movements. These workouts offer an effective approach to improving physical condition and sporting performance. They facilitate various physiological adaptations that favor the improvement of VO_2_ max [26]. This sport includes technically demanding exercises that require sustained power peaks over time, which can lead to high levels of neuromuscular fatigue and very high metabolic and perceived intensity values. At the same time, the incidence of injury in this type of exercise is high, especially in the shoulders, hips, and knees [27], as it has been shown that fatigue modifies the biodynamics of movement [28]. For this reason, it is important to pay special attention to fatigue during exercise.

Fatigue can be understood as a reduction in the ability to maintain power output stemming from various sources. Central fatigue arises from the CNS and is influenced by neurophysiological and psychological factors, such as metabolic changes, sensory input, and emotional state [29,30]. Peripheral fatigue, on the other hand, is related to the muscles and peripheral tissues, where physiological changes can affect the subjective feeling of fatigue [31,32]. Muscle fatigue specifically refers to the neuromuscular system’s reduced capacity to generate force, often due to metabolic byproducts like lactate. This affects muscle contraction and can alter mechanical outputs like strength and power [33,34]. Each type of fatigue interacts, but they do not completely overlap, meaning perceived fatigue can vary widely among individuals for the same physiological condition [35,36,37].

From a physiological perspective, high-intensity interval training (HIFT) primarily utilizes ATP resynthesis via phosphocreatine as the primary energy pathway [38,39]. The action of the enzyme creatine kinase represents a limiting factor in this process [40]. Phosphocreatine reserves are typically depleted within 10 to 30 s of HIFT [41]. Therefore, this depletion of phosphocreatine reserves is the primary factor in neuromuscular fatigue during this type of exercise, with the decrease in pH being less relevant in this variable [42,43].

HIFT is a widely recognized exercise modality known for its efficacy in improving cardiovascular fitness, metabolic health, and muscular endurance. However, HIIT can also lead to significant neuromuscular fatigue, which may impair exercise performance and recovery. β-Alanine, a non-essential amino acid, has been posited to enhance exercise performance by increasing muscle carnosine concentrations, which in turn may buffer the acidic environment resulting from high-intensity exercise. We hypothesize that β-alanine supplementation will attenuate neuromuscular fatigue and reduce metabolic and perceived exercise intensity in subjects performing HIIT. This attenuation is theorized to stem from β-alanine’s role in augmenting intramuscular buffering capacity through elevated carnosine levels, thereby delaying the onset of fatigue. The objective of this study is to assess the impact of 4-week supplementation with β-alanine on neuromuscular fatigue in individuals engaged in high-intensity interval training. This study will examine the effects on neuromuscular fatigue levels and perceived exercise intensity, as well as metabolic changes.

## 2. Materials and Methods

### 2.1. Experimental Design

The sample was selected by non-probability sampling according to the following inclusion criteria: being a non-sedentary subject, performing high-intensity interval exercise at least 3 times a week during the last 3 months. Being between 18 and 50 years of age. Patients with any type of disease that contraindicates the practice of exercise were excluded, such as CNS, metabolic, cardiovascular, pregnant women, etc., as well as those intolerant or allergic to b-alanine or sucrose, and subjects supplementing with proteins or other amino acids were also excluded.

This study involved a total of 27 subjects, who were divided into two groups (Group A as the control group and Group B as the experimental group), as shown in Figure 1. Group A was supplemented with Hacendado^®^ brand sucrose powder, while Group B was supplemented with 100% raw, HSN^®^ brand (Granada, Spain) β-alanine powder. Following the review of compliance with the inclusion criteria, the subjects were randomly assigned to one of the two groups. The randomization of participants into the experimental and control groups was achieved in a blinded manner with electronic data capture. A total of 27 subjects were selected for the study, with 13 assigned to the control group and 14 assigned to the experimental group. The participants were assembled and provided with comprehensive information regarding the specifics of the study. All of them voluntarily signed the informed consent form (Figure 1).

This study was conducted over four weeks, with both groups engaging in high-intensity interval training (HIFT) on the first day to induce fatigue and work at ranges close to their VO_2_ max. This was repeated four weeks after the β-alanine or sucrose intake protocol. The training was carried out on the same day of the week for both the pre- and post-measurement periods, with the same training also conducted on the day before to prevent the biasing of the results due to accumulated fatigue.

Before the completion of the Work of the Day (WOD), all subjects engaged in a warm-up routine. This consisted of a period of five minutes of rowing, followed by a series of exercises designed to enhance joint mobility. The warm-up concluded with a series of 6 reps each of burpees, box jumps, and bear crawls; 8 reps of slam balls (5–10 kg); and 10 reps each of in and out agility ladder and medicine ball on chest sit-ups (5–10 kg).

Subsequently, the subjects were weighed on a scale and performed three vertical jumps against movement on the platform, from which an average of height and power was taken.

After the WOD, between 30 s and the next 3 min, lactic acid was measured, the subject was asked about their perception of exertion, which was noted on the modified Borg scale, and the three jumps were repeated. Measurements were taken between 30 s and 3 min post-training to obtain adequate phosphocreatine levels for fatigue levels [29]. However, from the 4th minute onwards, these levels increase, which may bias the result of the measurement.

At the end of the training, each subject was given the appropriate supplementation according to randomization, and the dosage was explained in detail. Following a four-week interval, the same procedure was repeated, with the HIFT conducted on the same day of the week as in week one. This was conducted to prevent any potential bias in the results.

### 2.2. Subjects

The individuals, consisting of 4 females and 23 males (with an average age of 28.70 ± 6.44 years, a height of 174.21 ± 8.70 cm, a weight of 75.18 ± 10.97 kg, and a body mass index of 24.62 ± 1.86 kg/m^2^), engaged in an FFT training session at a fitness center specifically designed for this type of exercise regimen. The participants needed to possess over 18 months of experience in resistance training, which should include Olympic lifts and the use of free weights in their regular workout routines. All participants demonstrated proficiency in performing a power clean with a load of 50 kg for men and 35 kg for women, as well as completing 15 consecutive pull-ups. In the 3 months preceding the research, participants were strictly advised against the consumption of narcotics, psychotropic substances, stimulants, nutritional supplements, or any performance-enhancing drugs. Additionally, individuals with neurological, pulmonary, metabolic, or cardiovascular conditions, as well as any orthopedic restrictions that could impede their performance or the proper execution of the exercises, were excluded from the study.

No participant within the study cohort possessed elite athletic status. The determination of the sample size was established based on the findings of an initial investigation following the identical research protocol, which engaged 10 individuals majoring in sports science. This computation was conducted utilizing a significance level of α = 0.05 (denoting a 5% probability of type I error) and a power of 1 − β = 0.80 (equating to 80% power) while referencing outcomes from antecedent research works where the sample size matched or was smaller. The derived sample size amounted to 28 individuals proficient in strength training. Prior to the commencement of the research, all participants were briefed on the study’s structure, as well as the assessments and physical activities involved, before providing their documented consent. The research blueprint was formally sanctioned by the ethics committee of our institution, CEI-UCJC_05_22_HIFTMEDD, in compliance with the principles outlined in the Helsinki Declaration [44].

### 2.3. Supplementation

A total of 8000 size 1, 0.8-g capsules were used to encapsulate the supplements. The encapsulation used was a manual Capsunorm Fastlock 100 (Microcaya^®^, Bizkaia, Spain) for 100 capsules. To carry out this research project, it was necessary to consider the dosage of the supplement to be administered and the possible adverse effects that the subjects might suffer because of taking the supplement [8,45,46,47]. Plasma b-alanine levels recover approximately 3 h after ingestion, so we allowed at least 3 h between intakes. Between 30 and 40 min after ingestion, these levels have already fallen by 50% [1,5].

Therefore, according to recent studies on optimal supplementation to avoid the adverse effects of β-alanine intake, an ideal supplementation of about 0.8 g of β-alanine every 3–4 h has been proposed, giving a daily intake of about 4–5 g, or between 50 and 60 mg/kg/day [1,48]. This dosage of intermittent and gradual intake throughout the day has a greater effect on the retention of β-alanine, which is useful for carnosine synthesis, as it reduces the urinary excretion of this compound and does not pose significant health risks to individuals, and this is the optimal dose to produce effects on fatigue and therefore exercise performance without producing adverse effects [49].

The supplementation period lasts for four weeks, with subjects taking a total dose of 4 g of β-alanine or a placebo daily. The supplements are divided into five doses of 0.8 g to avoid the side effects previously described in various articles, such as itching, paresthesia, or numbness. Accordingly, the subject should take one capsule after a meal, as described in the literature (breakfast, mid-morning, lunch, mid-afternoon, and dinner) [3]. This option is recommended as it has been observed that there is no paresthesia when β-alanine is ingested and administered with food, reducing peak serum concentrations by 50% due to the process of slowing gastric emptying [1].

### 2.4. High-Intensity Functional Training Workout

Following our preview study [50], the workout routine starts with a 5 min general warm-up consisting of light to moderate intensity rowing, followed by 5 min stretching and joint movements. These mobility exercises are designed to cover the full range of movements for the main joint areas and are led by a sports science professional. After the general warm-up, there is a specific FFT warm-up, which includes 6 repetitions each of burpees, box jumps, and bear crawls, 8 repetitions of slam balls (5–10 kg), and 10 repetitions each of in and out agility ladder and medicine ball on chest sit-ups (5–10 kg).

After completing the general and specific warm-ups, participants proceeded with a “rounds for time” high-intensity workout. This workout consisted of 2 rounds of:-6 power cleans (50 kg for men, 35 kg for women)-10 objects over the shoulder (20 kg for men, 15 kg for women)-14 wall ball shots (9 kg for men, 7 kg for women)-18 shoulders to overhead dumbbell (DB) movements (20 kg for men, 10 kg for women)-200 m run

After completing these two rounds, participants continued with an additional two rounds of:-6 pull-ups-10 bodyweight squats-14 DB power snatches (20 kg for men, 10 kg for women)-18 box jumps (60 cm for men, 50 cm for women)-100 m run (Table 1).

The functional flexibility training (FFT) program, shown in Table 1, was implemented by the study participants based on the study by García-Fernández et al. [50]. The exercise descriptions were derived from the 2021 movement standards set by the iF3. The Functional Fitness Training (FFT) workout mentioned adheres to the movement standards established by the International Functional Fitness Federation (iF3) for the year 2019 (https://functionalfitness.sport/wp-content/uploads/2021/01/2021_iF3_Movement_Standards.pdf, accessed on 24 May 2021). These standards are detailed in a document that outlines the exercises authorized for iF3 events in 2021, aiming to standardize athletic tasks. To ensure adherence to these standards, two investigators with experience in FFT supervised each participant during the workout.

### 2.5. Blood Lactate

Before the commencement of the warm-up and immediately following the completion of the FFT workout, capillary blood samples (5 µL) were collected from the index finger to determine the concentration of lactate in the blood. This was achieved using a portable analyzer that had been previously validated and calibrated (Lactate Pro LT-1710, Arkray Factory Inc., KDK Corporation, Siga, Japan) [51,52].

### 2.6. Muscular Fatigue

The assessment of muscular fatigue in the legs is conducted through the measurement of vertical reaction forces during a countermovement jump (CMJ). The CMJ test is performed on a force platform, which is capable of measuring forces ranging from 0 to 10 kN with a sampling velocity of 0.5 kHz [53]. This specific setup involves a portable force platform of dimensions 92 × 92 × 12.5 cm, connected to a personal computer that utilizes specialized software to analyze the data. The Quattro Jump model 9290AD from Kistler Instruments (Wien, Austria), along with its accompanying software version 1.1.1.4, is used to record and quantify various variables related to the jump performance. The subjects perform a warm-up, followed by the FFT workout, which includes a series of jumps starting from a standing position with hands on hips and legs extended. The movement sequence involves an initial downward phase (eccentric action), followed by a forceful upward leap (concentric action), aiming for maximum jump height. The technique emphasizes knee extension in mid-air and toe-first ground contact upon landing. Consistency in posture and avoidance of lateral movements are crucial throughout the exercise. After the FFT workout, a cool-down period is observed before proceeding with the CMJ tests [54,55]. The vertical velocity of the center of mass (COM) is calculated using the impulse method, which involves integrating the ground reaction force (GRF) before the jump. The maximum take-off velocity is reached at the end of the jump’s concentric phase, and jump height is derived from this velocity and the force of gravity [56]. Power is determined from the force-time curve using the impulse–momentum principle, and relative power is the product of mean velocity and the vertical GRF component [57,58]. These measurements provide valuable data for understanding an athlete’s condition and performance capabilities [59]. In biomechanics, the analysis of a vertical jump can provide insights into muscular power and performance. The maximum take-off velocity (Vmax) is a critical factor, as it directly influences the jump height. The formula for calculating jump height from Vmax is given by height = ((Vmax)2/2 × 9.81)) [60]. Power output during the jump can be derived from the force-time curve without filtering, applying the impulse–momentum theorem. This theorem states that the change in momentum equals the impulse applied to the system. The mean relative power, expressed in watts per kilogram, is then calculated by multiplying the mean velocity by the vertical component of the ground reaction force. This metric is crucial for understanding the efficiency of the jump and the athlete’s explosive strength. Understanding these principles is essential for athletes looking to optimize their performance in activities requiring explosive power.

### 2.7. Rating of Perceived Exertion

Each participant was asked to rate their level of exertion on a Borg scale [61] from 6 to 20, with 6 representing “very, very light” exertion and 20 representing “maximum exertion.” The RPE was recorded at the levels of cardiorespiratory (cardio), muscular, and general immediately after the completion of the workout and at 3 min post-FFT [29,62]. To this end, each participant was requested to indicate with their finger on a scale of size DIN-A3 the perceived exertion required for the exercise in terms of cardiorespiratory effort (RPE cardio), muscular effort (RPE muscular), and in general terms (RPE general). Before commencing the FFTs, it was confirmed that each participant was aware of the meaning of each RPE. Consequently, for the RPE cardio, participants were instructed to consider their heart rate and the degree of breathlessness they were experiencing. For RPE muscular, participants were instructed to assess the degree of fatigue and discomfort experienced in their leg and arm muscles. To assess RPE in general, subjects were asked to provide a subjective evaluation of the overall perceived exertion of the workout. Participants were instructed to refrain from verbal descriptions and to indicate the appropriate point on the scale with their fingers.

### 2.8. Statistical Analysis

All statistical tests were performed using the software package SPSS version 29.0 (SPSS Inc., Chicago, IL, USA). The Shapiro–Wilk test was utilized to check the normal distribution of the data. All data are provided as their means (M), standard deviations (SD), and medians when corresponding.

All variables, except for the Perceived Exertion Scale, exhibited a normal and homogeneous distribution within the baseline data. Consequently, parametric tests were performed, specifically the Student’s *t*-test. The variable, which was not found to be within the normal range (*p* < 0.05), was analyzed with the Mann–Whitney U test. For intergroup differential analysis, the independent sample *t*-test was employed, while for those that did not meet normality, Mann–Whitney U was used.

For intra-group analysis, the related samples *t*-test was employed, whereas, in the event of non-homogeneity, the Wilcoxon test was utilized. Qualitative variables were analyzed using goodness-of-fit with chi-square. Bivariate correlations were analyzed using Pearson’s or Spearman’s correlation, as required by the data. A significance level of *p* < 0.05 was set at 95% confidence, a value universally considered adequate in biomedical research.

## 3. Results

Table 2 presents the baseline data of the subjects, which allows us to observe the pre-intervention jumps 1 and 2, pre-intervention power 1 and 2, pre-intervention lactic acid, and weight. The variable perceived exertion scale, as indicated above, exhibited a normal and homogeneous distribution, so it was analyzed individually, with the placebo group presenting a median of 16.80 ± 1.57 (SD) *p* = 0.043 * and for the β-alanine group 17.36 ± 1.50 *p* = 0.056.

Table 3 presents the descriptive statistical data, which does not show statistically significant changes for the proposed intergroup variable.

Table 4 presents the results of the ANOVA of the repeated measures derived from the Student’s *t*-test. It reveals that there are no statistically significant changes (*p* < 0.05) in the intergroup analysis of the variables “scale of perceived effort” and “lactic acid.”

## 4. Discussion

The results indicate that the only statistically significant difference is observed in the variable relating to jump height. Regarding jump power or metabolic intensity of exercise, a tendency towards improvement is observed, although this is not statistically significant. These changes are only observed in the group that was supplemented with β-alanine. In the placebo group, no significant changes were observed in any of the variables measured.

HIFT and other high-metabolic-intensity exercises result in a significant expenditure of energy substrates, which involve the utilization of supplementary metabolic pathways, including anaerobic glycolysis. These pathways lead to the accumulation of metabolic waste products, such as lactate or hydrogen ions, which modify the intracellular pH, reducing it, particularly when the intensity of the exercise exceeds the aerobic removal capacity [63]. The accumulation of these metabolites, therefore, results in an impairment of muscle contraction, which in turn leads to a loss of strength and contributes to the onset of fatigue. A reduction in pH resulting from the accumulation of metabolic waste impairs the release and reuptake of calcium from the sarcoplasmic reticulum, which directly affects the coupling of actin and myosin during muscle contraction and reduces the reuptake of phosphoryl creatine. In addition, it inhibits glycolysis and the release and reuptake of calcium from the sarcoplasmic reticulum, which directly influences the coupling of actin and myosin during muscle contraction. Furthermore, it inhibits the reuptake of phosphoryl creatine and glycolysis [64,65]. This is evidenced by an increase in the athlete’s perception of effort and a concomitant decrease in electromyographic activity relative to the onset of fatigue [66,67,68,69].

Following a four-week training period, the group that had been supplemented with β-alanine exhibited an improvement in jump height and a tendency towards greater jumping power. This improvement may be related to its antioxidant effects, its involvement in the protection of glycation, and its relationship with the generation of carnosine. These effects serve to buffer the drop in pH generated by the accumulation of metabolic waste, which may increase muscle performance. Consequently, the group supplemented with b-alanine would have been able to accumulate a greater load, volume, and intensity during the workouts performed, which would have influenced the performance observed in these variables [8,70]. This suggests that there has been an improvement in sports performance. This improvement may have been due to the physiological adaptations produced in the subjects during this period, as all the subjects trained at least three days a week. Some authors maintain that β-alanine supplementation could improve performance indirectly, as it has been demonstrated that when a dose of 3.2 g/day of β-alanine was ingested together with 10.5 g/day of creatine for 10 weeks, a positive effect was observed on the volume and intensity of strength training performed, as well as the intensity of the same and a decrease in fat mass. It should be noted that the study did not exclude participants who were taking creatine, which could have been a confounding factor [71]. Van Thienen et al. observed improvements in peak power and average power in a Wingate test following the performance of a test at varying intensities above and below the maximum lactate steady state. They obtained an ergogenic effect on performance in pre-fatigue situations with an 8-week protocol and an increasing dose of between 2 and 4 g of β-alanine [72].

Despite the described effects of β-alanine supplementation, no changes in perceived exertion were observed in our study after four weeks of intervention. This may be attributed to the fact that the participants’ performance of the WOD is at its maximum and that the subjective variable remains unaltered despite the physiological effects of the supplementation. However, there is a trend with respect to the objective variable of metabolic intensity of exercise, which seems to show a decrease only in the supplemented group. This could indicate a superior metabolic adaptation to effort, particularly in HIFT-type exercises. The observed decline in lactate levels may, in part, account for the enhanced jump height performance and the upward trajectory in jump power observed in the β-alanine supplemented group. It is acknowledged that the measurements were taken after four weeks of supplementation, which may not have been sufficient to objectively demonstrate a significant improvement in these variables. The evidence suggests that performance improved after the fourth week of supplementation, indicating that the population examined may not have benefited from the effects on sporting performance. Furthermore, the administered dose was 4 g/day, which is below the minimum recommended dosage for obtaining significant results, which is defined as 6.4 g/day. 

It can, therefore, be concluded that, as previously proposed by Bogdanis and Suzuki, a reduction in muscle pH does not affect the performance of HIFT subjects. There is a correlation in terms of performance between muscle carnosine and the type of muscle fiber since, in the exercises performed, type II fibers are more important than type I fibers. Therefore, fiber type predominates over pH levels [73]. Accordingly, following the findings of Hill published in 2007, it may be posited that the dosage was insufficient to elicit a more pronounced effect in this exercise regimen. However, in the context of activities such as walking and cycling, the four-week program demonstrated a 13% improvement in fatigue levels in the experimental group, with a further 3.2% increase observed after 10 weeks, in comparison to the control group, which exhibited no change [3].

### Limitations

The participants had to do a set of exercises regularly. The timing of the measurements during the week, coupled with the participants’ level of fatigue, made it difficult to carry out follow-up measurements. Consequently, follow-up measurements could not be performed. We must consider the potential for the exercise routine to influence the results. It was also a limitation that we could not keep the participants unaware of the details of the study. It is also important to acknowledge the limitations of our study, including the number of subjects included in the sample. One potential future direction for this research could be to extend the supplementation period to ascertain whether increasing the duration of supplementation would result in a more pronounced statistical difference between the study groups rather than merely an observed trend. Similarly, it would be beneficial to examine the impact of varying the dose administered to the subjects in the experimental group while ensuring that the dosage does not exceed a level that could potentially lead to adverse effects.

## 5. Conclusions

The results of this study indicate that supplementation of 4 g/d of β-alanine for four weeks in subjects performing high-intensity intervallic training (HIFT) does not produce statistically significant benefits for fatigue-related variables, neither in terms of metabolic intensity nor perceived exertion. However, it can be observed that there is a tendency for significance to present higher levels of performance. A four-week program of β-alanine intake with the dosage used, despite showing a tendency to improve vertical jump and power, has not shown statistically significant results.

## Figures and Tables

**Figure 1 nutrients-16-02340-f001:**
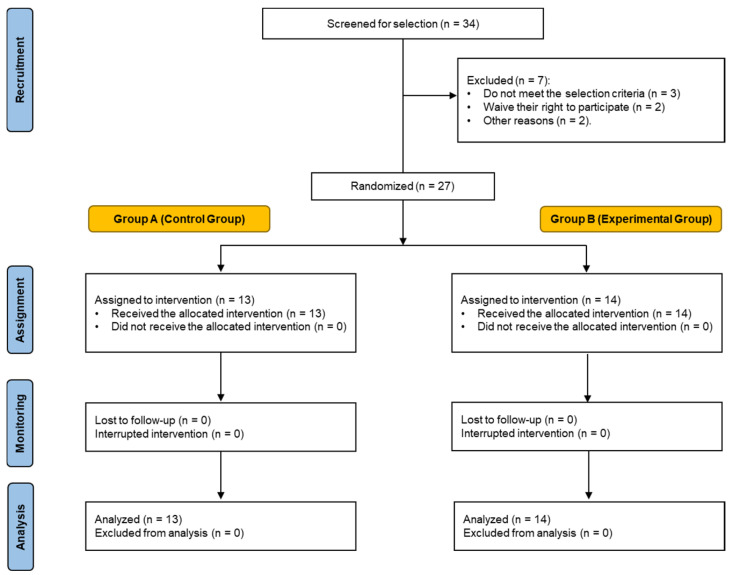
Sample flow chart.

**Table 1 nutrients-16-02340-t001:** Functional Fitness Training workout based on [50].

2 rounds (*r*) *×* (6 power clean)		2 r *×* (6 pull-up)	
** 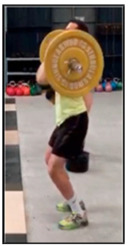 **	A power clean is a weightlifting exercise in which the athlete must catch the barbell in the front rack position without reaching the bottom of a squat.	** 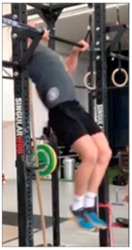 **	To complete the pull-up, the subject must grip a suspended horizontal bar with both hands and fully extend both arms at the repetition bottom. At the top of the repetition, the chin must break the uppermost horizontal plane of the bar. A kip of any style may be used. The repetition is counted when the athlete’s chin breaks the top-most horizontal plane of the bar.
Men: 50 kg	Women: 35 kg		
**2 r *×* (10 objects over shoulder)**		**2 r *×* (10 bodyweight squat)**	
** 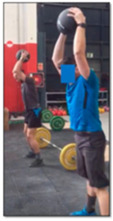 **	To perform the object-over-shoulder exercise, the athlete must first select the designated object and then lift it up and over their shoulder while simultaneously extending their hips fully. The athlete may employ any technique to lift the object.	** 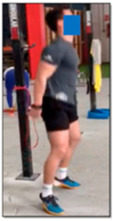 **	The bodyweight squat is a bodyweight movement in which the athlete begins in a standing position with an open hip angle, descends to a full squat with the creases of both hips below the plane of the top of the knees, and returns to the standing position with the hips returning to an open angle.
Men: 20 kg	Women: 15 kg		
**2 r *×* (14 wall ball shot)**			
** 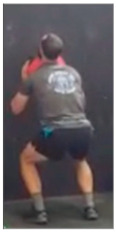 **	The wallball shot is executed with a medicine ball and an elevated target. The athlete must descend to a bottom-of-squat position with the medicine ball in the frontal plane, after which they must ascend while throwing the ball so that it makes contact at or above a designated height. While this repetition is permitted, it is not obligatory to jump during the execution.	** 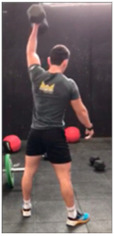 **	The power snatch is a weightlifting exercise in which the athlete must catch the dumbbell overhead with fully extended elbows without achieving the bottom of a squat during the task.
Men: 9 kg	Women: 7 kg	Men: 20 kg	Women: 10 kg
**2 r *×* (18 shoulder to overhead)**		**2 r *×* (18 box jump)**	
** 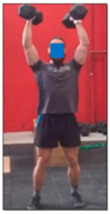 **	Shoulder-to-overhead movements entail the elevation of dumbbells from a static position at the shoulder to a static position overhead. The athlete may utilize a single, simultaneous flexion of the hips and/or knees to facilitate the elevation of the dumbbells to the top of the repetition.	** 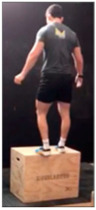 **	The box jump is a test of the athlete’s ability to initiate a jump from the ground with both feet simultaneously, land on a designated object (e.g., a box) with both feet, and demonstrate static control.
Men: 20 kg	Women: 10 kg	Men: 60 cm (23.6 inch)	Women: 50 cm (19.7 inch)
**2 r × (200 m run)**		**2 r × (100 m run)**	

**Table 2 nutrients-16-02340-t002:** Baseline data of the subjects.

	Placebo		β-Alanine		Levene	*p*-Valor
	M ± SD	Shapiro–Wilk	M ± SD	Shapiro–Wilk		
Pre-intervention Jump 1	29.37 ± 5.68	0.610	30.73 ± 6.51	0.210	0.882	0.551
Pre-intervention Power 1	937.19 ± 91.46	0.586	963.75 ± 117.16	0.459	0.701	0.500
Pre-intervencion Lactic acid	15.62 ± 5.34	0.395	17.58 ± 4.05	0.769	0.342	0.279
Pre-intervention Jump 2	28.78 ± 5.62	0.124	30.33 ± 6.37	0.201	0.983	0.490
Pre-intervention Power 2	926.95 ± 88.82	0.164	955.91 ± 110.87	0.151	0.755	0.443
Weight	72.40 ± 11.90	0.104	74,21 ± 10.61	0.151	0.256	0.669

M = mean ± SD = standard deviation.

**Table 3 nutrients-16-02340-t003:** Descriptive statistical data of the subjects.

		Baseline Pre-WOD 1	Baseline Post-WOD 2	Intervention Pre-WOD1	Intervention Post-WOD2	
		Mean ± DS	Mean ± DS	Mean ± DS	Mean ± DS	*p* Group × Time
Jump (cm)	Placebo	29.37 ± 5.68	28.78 ± 5.62	28.67 ± 5.14	28.60 ± 5.51	0.084
	β-alanine	30.73 ± 6.51	30.33 ± 6.37	31.09 ± 6.24	31.87 ± 5.46	
Power (Newtons)	Placebo	937.19 ± 91.46	926.95 ± 88.82	927.27 ± 83.19	925.21 ± 88.09	0.097
	β-alanine	963.75 ± 117.16	955.91 ± 110.87	963.51 ± 102.49	985.64 ± 92.40 †	

† = Significant pre-post WOD differences in the supplemented group post-intervention.

**Table 4 nutrients-16-02340-t004:** Results of the ANOVA of the repeated measures.

				Intra-Group Analysis		Inter-Group Analysis
		Pre-Intervention	Post-Intervention	Differences Pre-Post		
		Mean ± DS	Mean ± DS	Mean ± DS	*p*-Value	Student’s *t*-Test
EEP	Placebo	16.80 ± 1.57	17.40 ± 1.35	−0.6 ± 4.38	0.223	0.262
	β-alanine	17.36 ± 1.50	17.36 ± 1.39	0.00 ± 1.41	1.000	
Lactic acid (mmol·L^−1^)	Placebo	15.62 ± 5.34	17.06 ± 3.75	−1.44 ± 1.40	0.120	0.199
	β-alanine	17.58 ± 4.05	16.53 ± 5.09	1.05 ± 5.77	0.508	

## Data Availability

The original contributions presented in the study are included in the article, further inquiries can be directed to the corresponding author.

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
