# Peer review of "Effects of β-Alanine Supplementation on Subjects Performing High-Intensity Functional Training"

_nutrients, 2024, doi:10.3390/nu16142340_

Round 1

Reviewer 1 Report

Comments and Suggestions for Authors

Dear Authors,

The work requires editions, including:

1) The ‘Abstract’ should be changed in such a way that the ‘Introduction’ takes up less space and the methodology and results are described more precisely.

2) When using the HIFT abbreviation for the first time in the main text, please give the full form.

4) Do the examples of β-alanine supplementation in athletes given in the ‘Introduction’ section exhaust the literature review in this area?

5) Prior to providing the objective, it would be worth formulating a theoretical premise for undertaking such research, directly giving justification for studying this research problem.

6) II would suggest adding research questions and research hypotheses to the aim of the work.

7) Groups A and B can be clearly defined as control and study groups.

8) Please verify the number of respondents (27 or 29?).

9) Were all factors that could affect the results controlled as part of the experiment?

10) The abbreviation WOD should be given in full form if it appears for the first time in the main text.

11) Please, be careful not to repeat content in individual sections (and correct/eradicate existing repetitions).

12) In addition, please consider whether the excess of new paragraphs does not excessively atomise the text, e.g. it seems that there is no need to given a separate paragraph for central fatigue and peripheral fatigue in the text etc.

13) Please, explain what descriptive statistics are given in Table 2 (in the ‘Statistical analyses’ section, means are indicated, while in the table, medians are given?).

14) The paragraph on ‘Limitations’ is missing, which needs to be added.

15) In my opinion, the discussion is not thorough enough to consider the obtained results, both within the context of mechanisms and the research results achieved by other authors.

16) The conclusions should specify the group to which these results apply because they cannot be generalised.

17) I suggest analysing the literature in terms of publication date, including increasing the number of recent publications and limiting the number of publications from distant years.

Author Response

Response to reviewers' comments: Manuscript Nutrients-3099758 entitled “Effects of b-alanine supplementation in subjects performing high-intensity functional training."

We want to express our gratitude to the Journal Editor and the Reviewers for the time spent on our manuscript and for their helpful and constructive comments. 

We have addressed the points raised by the Reviewers in the response letter and changes have been highlighted (in red) in the manuscript. We believe the manuscript has been tuned in the light of the suggested additions.

Reviewer Comments to Author:

Reviewer 1:

The work requires editions including:

Comment 1:

1) The ‘Abstract’ should be changed in such a way that the ‘Introduction’ takes up less space and the methodology and results are described more precisely

The authors appreciate this comment. You are right. The “Abstract” was changed as requested.

Comment 2:

2) When using the HIFT abbreviation for the first time in the main text, please give the full form.

Thanks for this comment. You are right. We have included the full form of the abbreviation (line 70).

Comment 3:

3) Do the examples of β-alanine supplementation in athletes given in the ‘Introduction’ section exhaust the literature review in this area?

Thank you for pointing this out. The entire text has been revised to update the references as requested by the reviewer.

Comment 4:

4) Prior to providing the objective, it would be worth formulating a theoretical premise for undertaking such research, directly giving justification for studying this research problem.

Thank you for pointing this out. You are right. The theoretical premise for undertaking the research has been added as suggested by the reviewer.

Comment 5:

5) II would suggest adding research questions and research hypotheses to the aim of the work.

Thanks for this comment. You are right. The previous hypothesis has been added prior to the objectives as suggested by the reviewer.

Comment 6:

6) Groups A and B can be clearly defined as control and study groups.

Thanks for this comment. As indicated by the reviewer, both Group A (Control Group) and Group B (Experimental Group) have been described in the Abstract and in Material and Methods.

Comment 7:

7) Please verify the number of respondents (27 or 29?).

Thank you for your comments and the time devoted to correcting our study. The number of subjects (27) in our study has been revised and modified throughout the text.

Comment 8:

8) Were all factors that could affect the results controlled as part of the experiment?

The authors appreciate this comment. As suggested by the reviewer, an explanation of the different factors that could influence the results has been added.

Comment 9:

9) The abbreviation WOD should be given in full form if it appears for the first time in the main text.

Thanks for this comment. You are right. We have included the full form of the abbreviation

Comment 10:

10) Please, be careful not to repeat content in individual sections (and correct/eradicate existing repetitions).

Thanks for this comment. You are right. We have made the changes to the article as suggested by the reviewer.

Comment 11:

11) In addition, please consider whether the excess of new paragraphs does not excessively atomise the text, e.g. it seems that there is no need to given a separate paragraph for central fatigue and peripheral fatigue in the text etc.

Thanks for this comment. You are right. We have made the changes to the article as suggested by the reviewer.

Comment 12:

12) Please, explain what descriptive statistics are given in Table 2 (in the ‘Statistical analyses’ section, means are indicated, while in the table, medians are given?).

Thanks for this comment. You are right. We have made the changes to the article as suggested by the reviewer.

Comment 13:

13) The paragraph on ‘Limitations’ is missing, which needs to be added.

Thanks for this comment. You are right. We have included the “limitations” section as the reviewer suggested.

Comment 14:

14) In my opinion, the discussion is not thorough enough to consider the obtained results, both within the context of mechanisms and the research results achieved by other authors.

As the reviewer correctly notes, the discussion has been revised to facilitate a comparison of the findings of the present study with those of the existing literature.

Comment 15:

15) The conclusions should specify the group to which these results apply because they cannot be generalised.

Thanks for this comment. You are right. We have made the changes to the article as suggested by the reviewer.

Comment 16:

16) I suggest analysing the literature in terms of publication date, including increasing the number of recent publications and limiting the number of publications from distant years.

Thank you for pointing this out. The entire text has been revised to update the references as requested by the reviewer.

Reviewer 2 Report

Comments and Suggestions for Authors

The authors conducted a pilot randomized controlled pilot study to evaluate the effects of 4-week supplementation with β-alanine on neuromuscular fatigue in subjects who perform High-Intensity Functional Training (HIFT), as well as to examine the impact of this supplementation on sports performance. The authors report significant changes in some sports performance variables, specifically vertical jump and jumping power. These changes were observed only in the group that received supplementation. 

1) Please structure the manuscript as per CONSORT guidelines and please ensure all elements stated in CONSORT guidelines are covered in this manuscript. For more info, please see: https://www.mdpi.com/editorial_process#standards

For example, randomization process is not described in sufficient detail.

2) In the abstract, please mention that this is a randomized controlled trial and share more numerical details in abstract results. To make space, you could reduce the background section in the abstract.

3) The text states 27 subjects were included; but Figure 1 says 29 subjects were randomized and 30 subjects were analyzed- these numbers appear incorrect (for e.g., if you randomized 29 subjects- how did you analyze 30 subjects?); please correct these numbers and recheck the manuscript for accuracy of data.

4) Some text, for e.g. some text in table 5, is not in English language- please make sure entire manuscript is in English language.

5) Was multiple comparisons adjustment done for p-values- based on Bonferroni correction, there are likely no significant differences between the two trial arms. Please adjust the conclusions in the manuscript and in the abstract as well to avoid overinterpretations of findings.

6) Please add a limitations para at the end of the discussion section enlisting limitations of this study.

Comments on the Quality of English Language

n/a

Author Response

Response to reviewers' comments: Manuscript Nutrients-3099758 entitled “Effects of b-alanine supplementation in subjects performing high-intensity functional training."

We want to express our gratitude to the Journal Editor and the Reviewers for the time spent on our manuscript and for their helpful and constructive comments. 

We have addressed the points raised by the Reviewers in the response letter and changes have been highlighted (in red) in the manuscript. We believe the manuscript has been tuned in the light of the suggested additions.

Reviewer Comments to Author:

Reviewer 2:

The authors conducted a pilot randomized controlled pilot study to evaluate the effects of 4-week supplementation with β-alanine on neuromuscular fatigue in subjects who perform High-Intensity Functional Training (HIFT), as well as to examine the impact of this supplementation on sports performance. The authors report significant changes in some sports performance variables, specifically vertical jump and jumping power. These changes were observed only in the group that received supplementation.

Comment 1:

1) Please structure the manuscript as per CONSORT guidelines and please ensure all elements stated in CONSORT guidelines are covered in this manuscript. For more info, please see: https://www.mdpi.com/editorial_process#standards

For example, randomization process is not described in sufficient detail.

The authors appreciate this comment. In accordance with the reviewer's recommendation, the following elements have been incorporated into the manuscript: the hypothetical premise, the theoretical hypothesis, the rationale behind the randomization of the subjects, and an analysis of the potential confounding factors that could influence the results.

Comment 2:

2) In the abstract, please mention that this is a randomized controlled trial and share more numerical details in abstract results. To make space, you could reduce the background section in the abstract.

Thanks for this comment. You are right. Following your comment, the study has been enhanced by the inclusion of a randomized controlled clinical trial specification. The Abstract has been revised to provide a more coherent overview of the background and to expand the remaining sections. This was a recommendation made by the reviewer.

Comment 3:

3) The text states 27 subjects were included; but Figure 1 says 29 subjects were randomized and 30 subjects were analyzed- these numbers appear incorrect (for e.g., if you randomized 29 subjects- how did you analyze 30 subjects?); please correct these numbers and recheck the manuscript for accuracy of data.

Thank you for your comments and the time devoted to correcting our study. The number of subjects (27) in our study has been revised and modified throughout the text.

Comment 4:

4) Some text, for e.g. some text in table 5, is not in English language- please make sure entire manuscript is in English language.

Thank you for this comment. The language has been revised to ensure that it is exclusively in English.

Comment 5:

5) Was multiple comparisons adjustment done for p-values- based on Bonferroni correction, there are likely no significant differences between the two trial arms. Please adjust the conclusions in the manuscript and in the abstract as well to avoid overinterpretations of findings.

The authors appreciate this comment. Thanks for this comment. To facilitate a more comprehensive and coherent understanding of the material by the reader, the authors have consolidated the data from disparate tables, eliminating those that were deemed to be of insufficient relevance

Comment 5:

6) Please add a limitations para at the end of the discussion section enlisting limitations of this study.

Thanks for this comment. You are right. We have included the “limitations” section as the reviewer suggested.

Round 2

Reviewer 2 Report

Comments and Suggestions for Authors

No additional comments